# Zeroth-Order Hard-Thresholding: Gradient Error vs. Expansivity

**William de Vazelhes**[1], **Hualin Zhang**[2], **Huimin Wu**[2], **Xiao-Tong Yuan**[2], **Bin Gu**[1]
[1]Mohamed bin Zayed University of Artificial Intelligence
[2]Nanjing University of Information Science & Technology
{wdevazelhes,zhanghualin98,xtyuan1980,jsgubin}@gmail.com,
wuhuimin@nuist.edu.cn

## Abstract

$\ell_0$ constrained optimization is prevalent in machine learning, particularly for high-dimensional problems, because it is a fundamental approach to achieve sparse learning. Hard-thresholding gradient descent is a dominant technique to solve this problem. However, first-order gradients of the objective function may be either unavailable or expensive to calculate in a lot of real-world problems, where zeroth-order (ZO) gradients could be a good surrogate. Unfortunately, whether ZO gradients can work with the hard-thresholding operator is still an unsolved problem. To solve this puzzle, in this paper, we focus on the $\ell_0$ constrained black-box stochastic optimization problems, and propose a new stochastic zeroth-order gradient hard-thresholding (SZOHT) algorithm with a general ZO gradient estimator powered by a novel random support sampling. We provide the convergence analysis of SZOHT under standard assumptions. Importantly, we reveal a conflict between the deviation of ZO estimators and the expansivity of the hard-thresholding operator, and provide a theoretical minimal value of the number of random directions in ZO gradients. In addition, we find that the query complexity of SZOHT is independent or weakly dependent on the dimensionality under different settings. Finally, we illustrate the utility of our method on a portfolio optimization problem as well as black-box adversarial attacks.

## 1 Introduction

$\ell_0$ constrained optimization is prevalent in machine learning, particularly for high-dimensional problems, because it is a fundamental approach to achieve sparse learning. In addition to improving the memory, computational and environmental footprint of the models, these sparse constraints help reduce overfitting and obtain consistent statistical estimation [46, 5, 33, 29]. We formulate the problem as follows:

$$\min_{\boldsymbol{x} \in \mathbb{R}^d} \left\{ f(\boldsymbol{x}) := \mathbb{E}_{\boldsymbol{\xi}} f(\boldsymbol{x}, \boldsymbol{\xi}) \right\}, \quad \text{s.t.} \quad \|\boldsymbol{x}\|_0 \leq k \tag{1}$$

where $f(\cdot, \boldsymbol{\xi}) : \mathbb{R}^d \to \mathbb{R}$ is a differentiable function and $\boldsymbol{\xi}$ is a noise term, for instance related to an underlying finite sum structure in $f$, of the form: $\mathbb{E}_{\boldsymbol{\xi}} f(\boldsymbol{x}, \boldsymbol{\xi}) = \frac{1}{n} \sum_{i=1}^{n} f_i(\boldsymbol{x})$. Hard-thresholding gradient algorithm [17, 31, 45] is a dominant technique to solve this problem. It generally consists in alternating between a gradient step, and a hard-thresholding operation which only keeps the $k$-largest components (in absolute value) of the current iterate. The advantage of hard-thresholding over its convex relaxations ([39, 41]) is that it can often attain similar precision, but is more computationally efficient, since it can directly ensure a desired sparsity level instead of tuning an $\ell_1$ penalty or constraint. The only expensive computation in hard-thresholding is the hard-thresholding step itself, which requires finding the top $k$ elements of the current iterate. Hard-thresholding was originally developed in its full gradient form [17], but has been later on extended to the stochastic setting by

Table 1: Complexity of sparsity-enforcing algorithms. We give the query complexity for a precision $\varepsilon$, up to the system error (see section 4). For first-order algorithms (FO), we give it in terms of number of first order oracle calls (#IFO), that is, calls to $\nabla f(x, \boldsymbol{\xi})$, and for ZO algorithms, in terms of calls of $f(\boldsymbol{\xi}, \cdot)$. Here $\kappa$ denotes the condition number $\frac{L}{\nu}$, with $L$ is the smoothness (or RSS) constant and $\nu$ is the strong-convexity (or RSC) constant.

| Type | Name | Assumptions | #IZO/#IFO | #HT ops. |
|------|------|-------------|-----------|----------|
| FO/$\ell_0$ | StoIHT [31] | RSS, RSC | $\mathcal{O}(\kappa \log(\frac{1}{\varepsilon}))$ | $\mathcal{O}(\kappa \log(\frac{1}{\varepsilon}))$ |
| ZO/$\ell_1$ | RSPGF [14] | smooth[3] | $\mathcal{O}(\frac{d}{\varepsilon^2})$ | — |
| ZO/$\ell_1$ | ZSCG[2][2] | convex, smooth | $\mathcal{O}(\frac{d}{\varepsilon^2})$ | — |
| ZO/$\ell_1$ | ZORO [7] | $s$-sparse gradient, weakly sparse hessian, smooth[3] RSC$_{\text{bis}}$[1] | $\mathcal{O}(s\log(d)\log(\frac{1}{\varepsilon}))$ | — |
| ZO/$\ell_0$ | **SZOHT** | RSS, RSC | $\mathcal{O}((k+\frac{d}{s_2})\kappa^2\log(\frac{1}{\varepsilon}))$ | $\mathcal{O}(\kappa^2\log(\frac{1}{\varepsilon}))$ |
| ZO/$\ell_0$ | **SZOHT** | smooth, RSC | $\mathcal{O}(k\kappa^2\log(\frac{1}{\varepsilon}))$ | $\mathcal{O}(\kappa^2\log(\frac{1}{\varepsilon}))$ |

[1] The definition of Restricted Strong Convexity from [7] is different from ours and that of [31], hence the bis subscript.
[2] We refer to the modified version of ZSCG (Algorithm 3 in [2]).
[3] RSPGF and ZORO minimize $f(x) + \lambda\|x\|_1$: only $f$ needs to be smooth.

Nguyen et al. [31], which developed a stochastic gradient descent (SGD) version of hard thresholding (StoIHT), and further more with Zhou et al. [47], Shen and Li [36] and Li et al. [20], which used variance reduction technique to improve upon StoIHT.

However, the first-order gradients used in the above methods may be either unavailable or expensive to calculate in a lot of real-world problems. For example, in certain graphical modeling tasks [42], obtaining the gradient of the objective function is computationally hard. Even worse, in some settings, the gradient is inaccessible by nature, for instance in bandit problems [35], black-box adversarial attacks [40, 9, 10], or reinforcement learning [34, 27, 11]. To tackle those problems, ZO optimization methods have been developed [30]. Those methods usually replace the inaccessible gradient by its finite difference approximation which can be computed only from function evaluations, following the idea that for a differentiable function $f : \mathbb{R} \to \mathbb{R}$, we have: $f'(x) = \lim_{h \to 0} \frac{f(x+h)-f(x)}{h}$. Later on, ZO methods have been adapted to deal with a convex constraints set, and can therefore be used to solve the $\ell_1$ convex relaxation of problem (1). To that end, Ghadimi et al. [14], and Cai et al. [7] introduce proximal ZO algorithms, Liu et al. [24] introduce a ZO projected gradient algorithm and Balasubramanian and Ghadimi [2] introduce a ZO conditional gradient [19] algorithm. We provide a review of those results in Table 1. As can be seen from the table, their query complexity is high (linear in $d$), except [7] that has a complexity of $\mathcal{O}(s\log(d)\log(\frac{1}{\varepsilon}))$, but assumes that gradients are sparse. In addition, those methods must introduce a hyperparameter $\lambda$ (the strength of the $\ell_1$ penalty) or $R$ (the radius of the $\ell_1$ ball), which need to be tuned to find which value ensures the right sparsity level. Therefore, it would be interesting to use the hard-thresholding techniques described in the previous paragraph, instead of those convex relaxations.

Unfortunately, ZO hard-thresholding gradient algorithms have not been exploited formally. Even more, whether ZO gradients can work with the hard-thresholding operator is still an unknown problem. Although there was one related algorithm proposed recently by Balasubramanian and Ghadimi [2], they did not target $\ell_0$ constrained optimization problem and importantly have strong assumptions in their convergence analysis. Indeed, they assume that the gradients, as well as the solution of the unconstrained problem, are $s$-sparse: $\|\nabla f(\boldsymbol{x})\|_0 \leq s$ and $\|\boldsymbol{x}^*\|_0 \leq s^* \approx s$, where $\boldsymbol{x}^* = \arg\min_{\boldsymbol{x}} f(\boldsymbol{x})$. In addition, it was recently shown by Cai et al. [7] that they must in fact assume that the support of the gradient is fixed for all $\boldsymbol{x} \in \mathcal{X}$, for their convergence result to hold, which is a hard limitation, since that amounts to say that the function $f$ depends on $s$ fixed variables.

To fill this gap, in this paper, we focus on the $\ell_0$ constrained black-box stochastic optimization problems, and propose a novel stochastic zeroth-order gradient hard-thresholding (SZOHT) algorithm. Specifically, we propose a dimension friendly ZO gradient estimator powered by a novel random support sampling technique, and then embed it into the standard hard-thresholding operator.

We then provide the convergence and complexity analysis of SZOHT under the standard assumptions of sparse learning, which are restricted strong smoothness (RSS), and restricted strong convexity (RSC) [31, 36], to retain generality, therefore providing a positive answer to the question of whether ZO gradients can work with the hard-thresholding operator. Crucial to our analysis is to provide carefully tuned requirements on the parameters $q$ (the number of random directions used to estimate the gradient, further defined in Section 3.1) and $k$. Finally, we illustrate the utility of our method on a portfolio optimization problem as well as black-box adversarial attacks, by showing that our method can achieve competitive performance in comparison to state of the art methods for sparsity-enforcing zeroth-order algorithm described in Table 1, such as [14, 2, 7].

Importantly, we also show that in the smooth case, the query complexity of SZOHT is independent of the dimensionality, which is significantly different to the dimensionality dependent results for most existing ZO algorithms. Indeed, it is known from Jamieson et al. [18] that the worst case query complexity of ZO optimization over the class $\mathcal{F}_{\nu,L}$ of $\nu$-strongly convex and $L$-smooth functions defined over a convex set $\mathcal{X}$ is linear in $d$. Our work is thus in line with other works achieving dimension-insensitive query complexity in zeroth-order optimization such as [15, 37, 44, 7, 6, 2, 7, 22, 18], but contrary to those, instead of making further assumptions (i.e. restricting the class $\mathcal{F}_{\nu,L}$ to a smaller class), we bypass the impossibility result by replacing the convex feasible set $\mathcal{X}$ by a *non-convex* set (the $\ell_0$ ball), which is how we can avoid making stringent assumptions on the class of functions $f$.

**Contributions.** We summarize the main contributions of our paper as follows:

1. We propose a new algorithm SZOHT that is, up to our knowledge, the first zeroth-order sparsity constrained algorithm that is dimension independent under the smoothness assumption, without assuming any gradient sparsity.
2. We reveal an interesting conflict between the error from zeroth-order estimates and the hard-thresholding operator, which results in a minimal value for the number of random directions $q$ that is necessary to ensure at each iteration.
3. We also provide the convergence analysis of our algorithm in the more general RSS setting, providing, up to our knowledge, the first zeroth-order algorithm that can work with the usual assumptions of RSS/RSC from the hard-thresholding literature.

## 2 Preliminaries

Throughout this paper, we denote by $\|\boldsymbol{x}\|$ the Euclidean norm for a vector $\boldsymbol{x} \in \mathbb{R}^d$, by $\|\boldsymbol{x}\|_\infty$ the maximum absolute component of that vector, and by $\|\boldsymbol{x}\|_0$ the $\ell_0$ norm (which is not a proper norm). For simplicity, we denote $f_{\boldsymbol{\xi}}(\cdot) := f(\cdot, \boldsymbol{\xi})$. We call $\boldsymbol{u}_F$ (resp. $\nabla_F f(\boldsymbol{x})$) the vector which sets all coordinates $i \notin F$ of $\boldsymbol{u}$ (resp. $\nabla f(\boldsymbol{x})$) to 0. We also denote by $\boldsymbol{x}^*$ the solution of problem (1) defined in the introduction, for some target sparsity $k^*$ which could be smaller than $k$. To derive our result, we will need the following assumptions on $f$.

**Assumption 1** (($\nu_s, s$)-RSC, [17, 28, 26, 45, 20, 36, 31]). *$f$ is said to be $\nu_s$ restricted strongly convex with sparsity parameter $s$ if it is differentiable, and there exist a generic constant $\nu_s$ such that for all $(\boldsymbol{x}, \boldsymbol{y}) \in \mathbb{R}^d$ with $\|\boldsymbol{x} - \boldsymbol{y}\|_0 \leq s$:*

$$f(\boldsymbol{y}) \geq f(\boldsymbol{x}) + \langle \nabla f(\boldsymbol{x}), \boldsymbol{y} - \boldsymbol{x} \rangle + \frac{\nu_s}{2} \|\boldsymbol{x} - \boldsymbol{y}\|^2$$

**Assumption 2** (($L_s, s$)-RSS, [36, 31]). *For almost any $\boldsymbol{\xi}$, $f_{\boldsymbol{\xi}}$ is said to be $L_s$ restricted smooth with sparsity level $s$, if it is differentiable, and there exist a generic constant $L_s$ such that for all $(\boldsymbol{x}, \boldsymbol{y}) \in \mathbb{R}^d$ with $\|\boldsymbol{x} - \boldsymbol{y}\|_0 \leq s$:*

$$\|\nabla f_{\boldsymbol{\xi}}(\boldsymbol{x}) - \nabla f_{\boldsymbol{\xi}}(\boldsymbol{y})\| \leq L_s \|\boldsymbol{x} - \boldsymbol{y}\|$$

**Assumption 3** ($\sigma^2$-FGN [16], Assumption 2.3 (Finite Gradient Noise)). *$f$ is said to have $\sigma$-finite gradient noise if for almost any $\boldsymbol{\xi}$, $f_{\boldsymbol{\xi}}$ is differentiable and the gradient noise $\sigma = \sigma(f, \boldsymbol{\xi})$ defined below is finite:*

$$\sigma^2 = \mathbb{E}_{\boldsymbol{\xi}}[\|\nabla f_{\boldsymbol{\xi}}(\boldsymbol{x}^*)\|_\infty^2]$$

**Remark 1.** *Even though the original version of [16] uses the $\ell_2$ norm, we use the $\ell_\infty$ norm here, in order to give more insightful results in terms of $k$ and $d$, as is done classically in $\ell_0$ optimization, similarly to [47]. We also note that in [16], $\boldsymbol{x}^*$ denotes an unconstrained minimum when in our case it denotes the constrained minimum for some sparsity $k^*$.*

For Corollary 2, we will also need the more usual smoothness assumption:

**Assumption 4** (L-smooth). *For almost any $\boldsymbol{\xi}$, $f_{\boldsymbol{\xi}}$ is said to be L smooth, if it is differentiable, and for all $(\boldsymbol{x}, \boldsymbol{y}) \in \mathbb{R}^d$ :*

$$\|\nabla f_{\boldsymbol{\xi}}(\boldsymbol{x}) - \nabla f_{\boldsymbol{\xi}}(\boldsymbol{y})\| \leq L\|\boldsymbol{x} - \boldsymbol{y}\|$$

## 3 Algorithm

### 3.1 Random support Zeroth-Order estimate

In this section, we describe our zeroth-order gradient estimator. It is basically composed of a random support sampling step, followed by a random direction with uniform smoothing on the sphere supported by this support. We also use the technique of averaging our estimator over $q$ dimensions, as described in [25]. More formally, our gradient estimator is described below:

$$\hat{\nabla} f_{\boldsymbol{\xi}}(\boldsymbol{x}) = \frac{d}{q\mu} \sum_{i=1}^{q} \left( f_{\boldsymbol{\xi}}(\boldsymbol{x} + \mu \boldsymbol{u}_i) - f_{\boldsymbol{\xi}}(\boldsymbol{x}) \right) \boldsymbol{u}_i \tag{2}$$

where each random direction $\boldsymbol{u}_i$ is a unit vector sampled uniformly from the set $\mathcal{S}_{s_2}^d := \{\boldsymbol{u} \in \mathbb{R}^d : \|\boldsymbol{u}\|_0 \leq s_2, \|\boldsymbol{u}\| = 1\}$. We can obtain such vectors $\boldsymbol{u}$ by sampling first a random support $S$ (i.e. a set of coordinates) of size $s_2$ from $[d]$, (denoted as $S \sim \mathcal{U}(\binom{[d]}{s_2})$) in Algorithm 1) and then by sampling a random unit vector $\boldsymbol{u}$ supported on that support $S$, that is, uniformly sampled from the set $\mathcal{S}_S^d := \{\boldsymbol{u} \in \mathbb{R}^d : \boldsymbol{u}_{[d]-S} = \boldsymbol{0}, \|\boldsymbol{u}\| = 1\}$, (denoted as $\boldsymbol{u} \sim \mathcal{U}(\mathcal{S}_S^d)$ in algorithm 1). The original uniform smoothing technique on the sphere is described in more detail in [12]. However, in our case, the sphere along which we sample is restricted to a random support of size $s_2$. Our general estimator, through the setting of the variable $s_2$, can take several forms, which are similar to pre-existing gradient estimators from the literature described below:

- If $s_2 = d$, $\hat{\nabla} f_{\boldsymbol{\xi}}(\boldsymbol{x})$ is the *usual vanilla estimator with uniform smoothing on the sphere* [12].
- If $1 \leq s_2 \leq d$, our estimator is similar to the Random Block-Coordinate gradient estimator from Lian et al. [21], except that the blocks are not fixed at initialization but chosen randomly, and that we use a uniform smoothing with forward difference on the given block instead of a coordinate-wise estimation with central difference. This random support technique allows us to give a convergence analysis under the classical assumptions of the hard-thresholding literature (see Remark 3), and to deal with huge scale optimization, when sampling uniformly from a unit $d$-sphere is costly [7, 6]: in the distributed setting for instance, each worker would just need to sample an $s_2$-sparse random vector, and only the centralized server would materialize the full gradient approximation containing up to $qs_2$ non-zero entries.

**Error Bounds of the Zeroth-Order Estimator.** We now derive error bounds on the gradient estimator, that will be useful in the convergence rate proof, except that we consider *only the restriction to some support $F$* (that is, we consider a subset of components of the gradient/estimator). Indeed, proofs in the hard-thresholding literature (see for instance [45]), are usually written only on that support. That is the key idea which explains how the dimensionality dependence is reduced when doing SZOHT compared to vanilla ZO optimization. We give more insight on the shape of the original distribution of gradient estimators, and the distribution of their projection onto a hyperplane $F$ in Figure 5 in Appendix E. We can observe that even if the original gradient estimator is poor, in the projected space, the estimation error is reduced, which we quantify in the proposition below.

**Proposition 1.** *(Proof in Appendix C.3 ) Let us consider any support $F \subset [d]$ of size $s$ ($|F| = s$). For the Z0 gradient estimator in (2), with $q$ random directions, and random supports of size $s_2$, and assuming that each $f_{\boldsymbol{\xi}}$ is $(L_{s_2}, s_2)$-RSS, we have, with $\hat{\nabla}_F f_{\boldsymbol{\xi}}(\boldsymbol{x})$ denoting the hard thresholding of the gradient $\nabla f_{\boldsymbol{\xi}}(\boldsymbol{x})$ on $F$ (that is, we set all coordinates not in $F$ to 0):*

(a) $\|\mathbb{E}\hat{\nabla}_F f_{\boldsymbol{\xi}}(\boldsymbol{x}) - \nabla_F f_{\boldsymbol{\xi}}(\boldsymbol{x})\|^2 \leq \varepsilon_{\mu} \mu^2$

(b) $\mathbb{E}\|\hat{\nabla}_F f_{\boldsymbol{\xi}}(\boldsymbol{x})\|^2 \leq \varepsilon_F \|\nabla_F f_{\boldsymbol{\xi}}(\boldsymbol{x})\|^2 + \varepsilon_{F^c} \|\nabla_{F^c} f_{\boldsymbol{\xi}}(\boldsymbol{x})\|^2 + \varepsilon_{abs} \mu^2$

(c) $\mathbb{E}\|\hat{\nabla}_F f_{\boldsymbol{\xi}}(\boldsymbol{x}) - \nabla_F f_{\boldsymbol{\xi}}(\boldsymbol{x})\|^2 \leq 2(\varepsilon_F + 1)\|\nabla_F f_{\boldsymbol{\xi}}(\boldsymbol{x})\|^2 + 2\varepsilon_{F^c} \|\nabla_{F^c} f_{\boldsymbol{\xi}}(\boldsymbol{x})\|^2 + 2\varepsilon_{abs} \mu^2$

$$\text{with} \quad \varepsilon_\mu = L_{s_2}^2 sd, \quad \varepsilon_F = \frac{2d}{q(s_2+2)}\left(\frac{(s-1)(s_2-1)}{d-1}+3\right)+2,$$

$$\varepsilon_{F^c} = \frac{2d}{q(s_2+2)}\left(\frac{s(s_2-1)}{d-1}\right) \text{ and } \varepsilon_{abs} = \frac{2dL_{s_2}^2 ss_2}{q}\left(\frac{(s-1)(s_2-1)}{d-1}+1\right)+L_{s_2}^2 sd \tag{3}$$

## 3.2 SZOHT Algorithm

We now present our full algorithm to optimize problem 1, which we name SZOHT (Stochastic Zeroth-Order Hard Thresholding). Each iteration of our algorithm is composed of two steps: (i) the gradient estimation step, and (ii) the hard thresholding step, where the gradient estimation step is the one described in the section above, and the hard-thresholding is described in more detail in the following paragraph. We give the full formal description of our algorithm in Algorithm 1.

In the hard thresholding step, we only keep the $k$ largest (in magnitude) components of the current iterate $x^t$. This ensures that all our iterates (including the last one) are $k$-sparse. This hard-thresholding operator has been studied for instance in [36], and possesses several interesting properties. Firstly, it can be seen as a projection on the $\ell_0$ ball. Second, importantly, it is not non-expansive, contrary to other operators like the soft-thresholding operator [36]. That expansivity plays an important role in the analysis of our algorithm, as we will see later.

Compared to previous works, our algorithm can be seen as a variant of Stochastic Hard Thresholding (StoIHT from [31]), where we replaced the true gradient of $f_\xi$ by the estimator $\hat{\nabla} f_\xi(x)$. It is also very close to Algorithm 5 from Balasubramanian and Ghadimi [2] (Truncated-ZSGD), with just a different zeroth-order gradient estimator: we use a uniform smoothing, random-block estimator, instead of their gaussian smoothing, full support vanilla estimator. This allows us to deal with very large dimensionalities, in the order of millions, similarly to Cai et al. [6]. Furthermore, as described in the Introduction, contrary to Balasubramanian and Ghadimi [2], we provide the analysis of our algorithm without any gradient sparsity assumption.

The key challenge arising in our analysis is described in Figure 1: the hard-thresholding operator being expansive [36], each approximate gradient step must approach the solution enough to stay close to it even after hard-thresholding. Therefore, it is *a priori* unclear whether the zeroth-order estimate can be accurate enough to guarantee the convergence of SZOHT. Hopefully, as we will see in the next section, we can indeed ensure convergence, as long as we carefully choose the value of $q$.

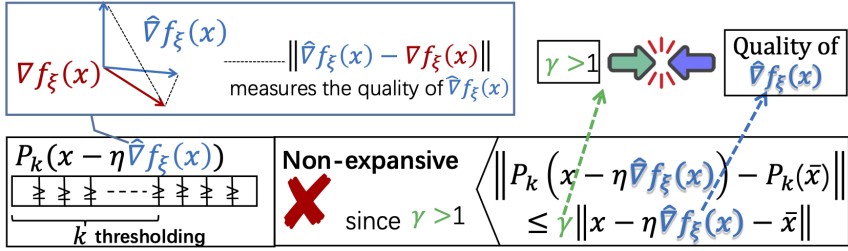

Figure 1: Conflict between the hard-thresholding operator and the zeroth-order estimate.

## 4 Convergence analysis

In this section, we provide the convergence analysis of SZOHT, using the assumptions from section 2, and discuss an interesting property of the combination of the zeroth-order gradient estimate and the hard-thresholding operator, providing a positive answer to the question from the previous section.

**Theorem 1.** *(Proof in Appendix D.1) Assume that that each $f_{\boldsymbol{\xi}}$ is $(L_{s'}, s' := \max(s_2, s))$-RSS, and that $f$ is $(\nu_s, s)$-RSC and $\sigma$-FGN, with $s = 2k + k^* \le d$, with $\frac{d-k^*}{2} \ge k \ge \rho^2 k^*/(1-\rho^2)^2$, with $\rho$ defined as below. Suppose that we run SZOHT with random supports of size $s_2$, $q$ random directions, a learning rate of $\eta = \frac{\nu_s}{(4\varepsilon_F+1)L_{s'}^2}$, and $k$ coordinates kept at each iterations. Then, we have a geometric convergence rate, of the following form, with $\boldsymbol{x}^{(t)}$ denoting the $t$-iterate of SZOHT:*

$$\mathbb{E}\|\boldsymbol{x}^{(t)} - \boldsymbol{x}^*\| \le (\gamma\rho)^t \|\boldsymbol{x}^{(0)} - \boldsymbol{x}^*\| + \left(\frac{\gamma a}{1-\gamma\rho}\right)\sigma + \left(\frac{\gamma b}{1-\gamma\rho}\right)\mu$$

**Algorithm 1:** Stochastic Zeroth-Order Hard-Thresholding (SZOHT)

---

**Initialization:** *Learning rate $\eta$, maximum number of iterations $T$, size of the random directions support $s_2$, number of random directions $q$, number of coordinates to keep at each iteration $k = \mathcal{O}(\kappa^4 k^*)$, initial point $\boldsymbol{x}^{(0)}$ with $\|\boldsymbol{x}^{(0)}\|_0 \leq k^*$ (typically $\boldsymbol{x}^{(0)} = 0$), .*

**Output:** $\boldsymbol{x}^T$.

**for** $t = 1, ..., T$ **do**

  Sample $\boldsymbol{\xi}$ (for instance sample a train sample $i$)

  **for** $i = 1, ..., q$ **do**

    Sample a random support $S \sim \mathcal{U}\left(\binom{[d]}{s_2}\right)$

    Sample a random direction $\boldsymbol{u}_i$ from the unit sphere supported on $S$: $\boldsymbol{u}_i \sim \mathcal{U}\left(\mathcal{S}_S^d\right)$

    Compute $\hat{\nabla} f_{\boldsymbol{\xi}}(\boldsymbol{x}^{t-1}; \boldsymbol{u}_i) = \frac{d}{\mu}\left(f_{\boldsymbol{\xi}}(\boldsymbol{x} + \mu \boldsymbol{u}_i) - f_{\boldsymbol{\xi}}(\boldsymbol{x})\right)\boldsymbol{u}_i$;

  **end**

  Compute $\hat{\nabla} f_{\boldsymbol{\xi}}(\boldsymbol{x}^{t-1}) = \frac{1}{q}\sum_{i=1}^{q}\hat{\nabla} f_{\boldsymbol{\xi}}(\boldsymbol{x}^{t-1}; \boldsymbol{u}_j)$

  Compute $\tilde{\boldsymbol{x}}^t = \boldsymbol{x}^{t-1} - \eta\hat{\nabla} f_{\boldsymbol{\xi}}(\boldsymbol{x}^{t-1})$;

  Compute $\boldsymbol{x}^t = \tilde{\boldsymbol{x}}_k^t$ as the truncation of $\tilde{\boldsymbol{x}}^t$ with top $k$ entries preserved;

**end**

---

$$\text{with} \quad a = \eta\left(\sqrt{(4\varepsilon_F s + 2) + \varepsilon_{F^c}(d-k)} + \sqrt{s}\right), \; b = \left(\frac{\sqrt{\varepsilon_\mu}}{L_{s'}} + \eta\sqrt{2\varepsilon_{abs}}\right),$$

$$\rho^2 = 1 - \frac{\nu_s^2}{(4\varepsilon_F + 1)L_{s'}^2}, \; \text{and} \; \gamma = \sqrt{1 + \left(k^*/k + \sqrt{(4 + k^*/k)\,k^*/k}\right)/2} \tag{4}$$

$$\text{and} \quad \varepsilon_F, \varepsilon_{abs}, \text{ and } \varepsilon_\mu \text{ are defined in (3)}.$$

**Remark 2** (System error). *The format of our result is similar to the ones in [45] and [31], in that it contains a linear convergence term, and a system error which depends on the expected norm of the gradient at $\boldsymbol{x}^*$ (through the variable $\sigma$). We note that if $f$ has a $k^*$-sparse unconstrained minimizer, which could happen in sparse reconstruction, or with overparameterized deep networks (see for instance [32, Assumption (2)]), then we would have $\|\nabla f(\boldsymbol{x}^*)\| = 0$, and hence that part of the system error would vanish. In addition to that usual system error, we also have here another system error, which depends on the smoothing radius $\mu$, due to the error from the ZO estimate.*

**Remark 3** (Generality). *If we take $s_2 \leq s$, the first assumption of Theorem 1 becomes the requirement that $f_{\boldsymbol{\xi}}$ is $(L_s, s)$-RSS. Therefore, SZOHT as well as the theorem above provides, up to our knowledge, the first algorithm that can work in the usual setting of hard-thresholding algorithms (that is, $(L_s, s)$-RSS and $(\nu_s, s)$-RSC [31, 36]), as well as its convergence rate.*

**Interplay between hard-thresholding and zeroth-order error** Importantly, contrary to previous works in ZO optimization, $q$ must be chosen carefully here, due to our specific setting combining ZO and hard-thresholding. Indeed, as described in [36], the hard-thresholding operator is not non-expansive (contrary to projection onto the $\ell_1$ ball) so it can drive the iterates away from the solution. Therefore, enough descent must be made by the (approximate) gradient step to get close enough to the solution, and it is therefore crucial to limit errors in gradient estimation. This problem arises with any kind of gradient errors: for instance with SGD errors [31, 47], it is generally dealt with either by ensuring some conditions on the function $f$ [31], forming bigger batches of samples [47], and/or considering a larger number of components $k$ kept in hard-thresholding (to make the hard-thresholding less expansive). In our work, similarly to Zhou et al. [47], we deal with this problem by relaxing $k$ and sampling more directions $\boldsymbol{u}_i$ (which is the ZO equivalent to taking bigger batch-size in SGD). However, there is an additional effect that happens in our case, specific to ZO estimation: as described in Proposition 1, the quality of our estimator *also depends on $k$*. Therefore, it may be hard to make the algorithm converge only by considering larger $k$: *higher $k$ means less expansivity (which helps convergence), but worse gradient estimate (which harms convergence)*. We further illustrate this conflict between the non-expansiveness of hard-thresholding (quantified by the parameter $\gamma$ [36]), and the error from the zeroth-order estimate, in Figure 1. Therefore, it is even more crucial to tune precisely our remaining degree of freedom at hand which is $q$. More precisely, a minimal value of $q$ is always necessary to ensure convergence in our setting, contrary to most ZO setting (in which

taking even $q = 1$ can work, as long as other constants like $\eta$ are well chosen, see for instance [23, Corollary 3]). The remark below gives some necessary conditions on $q$ to illustrate that fact.

**Remark 4** (Some necessary condition on q, proof in D.2). *Let $k^* \in \mathbb{N}^*$ and assume, that $k$ is such that $k > \rho^2 k^*/(1-\rho^2)^2$ (which ensures that $\rho\gamma < 1$), and that $k \leq \frac{d-k^*}{2}$. These conditions imply the following necessary (but not sufficient) condition on q:*

- *if $s_2 > 1$: $q \geq \frac{16d(s_2-1)k^*\kappa^2}{(s_2+2)(d-1)} \left[ 18\kappa^2 - 1 + 2\sqrt{9\kappa^2(9\kappa^2-1) + \frac{1}{2} - \frac{1}{2k^*} + \frac{3}{2}\frac{d-1}{k^*(s_2-1)}} \right]$*

- *if $s_2 = 1$: $q \geq \frac{8\kappa^2 d}{\sqrt{\frac{d}{k^*}+1}}$*

Remark 4 is just a warning that usual rules from ZO do not apply to SZOHT, but it does not say how to choose $q$ to ensure convergence: for that we would need some sufficient conditions on $q$ for Theorem 1 to apply. We give such conditions in the next section.

## 4.1 Weak/non dependence on dimensionality of the query complexity.

In this section, we provide Corollaries 1 and 2, following from Theorem 1, which give an example of $q$ that is sufficient to converge (that is, to obtain $\gamma\rho < 1$ in Theorem 1), and that achieves weak dimensionality dependence in the case of RSS, and complete dimension independence in the case of smoothness.

**Corollary 1** (RSS $f_{\boldsymbol{\xi}}$, proof in Appendix D.3). *Assume that that almost all $f_{\boldsymbol{\xi}}$ are $(L_{s'}, s' := \max(s_2, s))$-RSS, and that $f$ is $(\nu_s, s)$-RSC and $\sigma$-FGN, with $s = 2k + k^* \leq d$, with $\frac{d-k^*}{2} \geq k \geq (86\kappa^4 - 12\kappa^2)k^*$ (with $\kappa := \frac{L_{s'}}{\nu_s}$). Suppose that we run SZOHT with random support of size $s_2$, a learning rate of $\eta = \frac{\nu_s}{13L_{s'}^2}$, with $k$ coordinates kept at each iterations by the hard-thresholding, and with $q \geq 2s + 6\frac{d}{s_2}$. Then, we have a geometric convergence rate, of the following form, with $\boldsymbol{x}^{(t)}$ denoting the $t$-iterate of SZOHT:*

$$\mathbb{E}\|\boldsymbol{x}^{(t)} - \boldsymbol{x}^*\| \leq (\gamma\rho)^t \|\boldsymbol{x}^{(0)} - \boldsymbol{x}^*\| + \left(\frac{\gamma a}{1-\gamma\rho}\right)\sigma + \left(\frac{\gamma b}{1-\gamma\rho}\right)\mu$$

*with a, b and $\gamma$ are defined in (4), and $\rho = \sqrt{1 - \frac{2}{13\kappa^2}}$. Therefore, the query complexity (QC) to ensure that $\mathbb{E}\|\boldsymbol{x}^{(t)} - \boldsymbol{x}^*\| \leq \varepsilon + \left(\frac{\gamma a}{1-\gamma\rho}\right)\sigma + \left(\frac{\gamma b}{1-\gamma\rho}\right)\mu$ is $\mathcal{O}(\kappa^2(k + \frac{d}{s_2})\log(\frac{1}{\varepsilon}))$.*

We now turn to the case where the functions $f_{\boldsymbol{\xi}}$ are smooth. The key result in that case is that we can have a query complexity independent of the dimension $d$, which is, up to our knowledge, the first result of such kind for sparse zeroth-order optimization without assuming any gradient sparsity.

**Corollary 2** (Smooth $f_{\boldsymbol{\xi}}$, proof in Appendix D.4). *Assume that, in addition to the conditions from Corollary 1 above, almost all $f_{\boldsymbol{\xi}}$ are $L$-smooth, with $\frac{d-k^*}{2} \geq k \geq (86\kappa^4 - 12\kappa^2)k^*$ (with $\kappa := \frac{L}{\nu_s}$), and take $q \geq 2(s+2)$, and $s_2 = d$ (that is, no random support sampling). Then, we have a geometric convergence rate, of the following form, with $\boldsymbol{x}^{(t)}$ denoting the $t$-iterate of SZOHT:*

$$\mathbb{E}\|\boldsymbol{x}^{(t)} - \boldsymbol{x}^*\| \leq (\gamma\rho)^t \|\boldsymbol{x}^{(0)} - \boldsymbol{x}^*\| + \left(\frac{\gamma a}{1-\gamma\rho}\right)\sigma + \left(\frac{\gamma b}{1-\gamma\rho}\right)\mu$$

*Therefore, the QC to ensure that $\mathbb{E}\|\boldsymbol{x}^{(t)} - \boldsymbol{x}^*\| \leq \varepsilon + \left(\frac{\gamma a}{1-\gamma\rho}\right)\sigma + \left(\frac{\gamma b}{1-\gamma\rho}\right)\mu$ is $\mathcal{O}(\kappa^2 k \log(\frac{1}{\varepsilon}))$.*

Additionally, our convergence rate highlights an interesting connection between the geometry of $f$ (defined by the condition number $\kappa = L_{s'}/\nu_s$), and the number of random directions that we need to take at each iteration: if the problem is ill-conditioned, that is $\kappa$ is high, then we need a bigger $k$. This result is standard in the $\ell_0$ litterature (see for instance [45]). But specifically, in our ZO case, it also impacts the query complexity: since the projected gradient is harder to approximate when the dimension $k$ of the projection is larger, $q$ needs to grow too, resulting in higher query complexity. We believe this is an interesting result for the sparse zeroth-order optimization community: it reveals that the query complexity may in fact depend on some notion of intrinsic dimension to the problem, related to both the sparsity of the iterates $k$, and the geometry of the function $f$ for a given $s_2$ (through the restricted condition number $\kappa$), rather than the dimension of the original space $d$ as in previous works like [14].

# 5 Experiments

## 5.1 Sensitivity analysis

We first conduct a sensitivity parameter analysis on a toy example, to highlight the importance of the choice of $q$, as discussed in Section 4. We fix a target sparsity $k^* = 5$, choose $k = 74k^*$, and consider a sparse quadric function $f : \mathbb{R}^{5000} \to \mathbb{R}$, with: $f(\boldsymbol{x}) = \frac{1}{2}\|\boldsymbol{a} \odot (\boldsymbol{x} - \boldsymbol{b})\|^2$ ($\odot$ denotes the elementwise product), with $\boldsymbol{a}_i = 1$ if $i \geq d - s$ and $0$ otherwise (to ensure $f$ is $s$-RSC and smooth, with $\nu_s = L = 1$), and $\boldsymbol{b}_i = \frac{i}{100d}$ for all $i \in [d - 70k^*]$ and $0$ for all $d - 70k^* \leq i \leq d$ (we make such a choice in order to have $\|\nabla f(\boldsymbol{x}^*)\|$ small enough). We choose $\eta$ as in Theorem 1: $\eta = \frac{1}{(4\varepsilon_F + 1)}$ with $\varepsilon_F$ defined in Proposition 1 in terms of $s$ and $d$ (we take $s_2 = d$), $\mu = 1e - 4$, and present our results in Figure 2, for six values of $q$. We can observe on Figure 2(b) that the smaller the $q$, the less $f(\boldsymbol{x})$ can descend. Interestingly, we can also see on Figure 2(a) that for $q = 1$ and $20$, $\|\boldsymbol{x}^{(t)} - \boldsymbol{x}^*\|$ diverges: we can indeed compute that $\rho\gamma > 1$ for those $q$, which explains the divergence, from Theorem 1.

## 5.2 Baselines

We compare our SZOHT algorithms with state of the art zeroth-order algorithms that can deal with sparsity constraints, that appear in Table 1:

1. **ZSCG** [2] is a Frank-Wolfe ZO algorithm, for which we consider an $\ell_1$ ball constraint.

2. **RSPGF** [14] is a proximal ZO algorithm, for which we consider an $\ell_1$ penalty.

3. **ZORO** [7] is a proximal ZO algorithm, that makes use of sparsity of gradients assumptions, using a sparse reconstruction algorithm at each iteration to reconstruct the gradient from a few measurements. Similarly, as for ZSCG, we consider an $\ell_1$ penalty.

In all the applications below, we will tune the sparsity $k$ of SZOHT, the penalty of RSPGF and ZORO, and the radius of the constraint of ZSCG, such that all algorithms attain a similar converged objective value, for fair comparison.

## 5.3 Applications

We compare the algorithms above on two tasks: a sparse asset risk management task from [8], and an adversarial attack task [9] with a sparsity constraint.

**Sparse asset risk management**   We consider the portfolio management task and dataset from [8], similarly to [7]. We have a given portfolio of $d$ assets, with each asset $i$ giving an expected return $\boldsymbol{m}_i$, and with a global covariance matrix of the return of assets denoted as $\boldsymbol{C}$. The cost function we minimize is the portfolio risk: $\frac{\boldsymbol{x}^T \boldsymbol{C} \boldsymbol{x}}{2(\sum_{i=1}^d \boldsymbol{x}_i)^2}$, where $\boldsymbol{x}$ is a vector where each component $\boldsymbol{x}_i$ denotes how much is invested in each asset, and we require to minimize it under a constraint of minimal return $r$: $\frac{\sum_{i=1}^d \boldsymbol{m}_i \boldsymbol{x}_i}{\sum_{i=1}^d \boldsymbol{x}_i}$. We enforce that constraint using the Lagrangian form below. Finally, we add a sparsity constraint, to restrict the investments to only $k$ assets. Therefore, we obtain the cost function below:

$$\min_{x \in \mathbb{R}^d} \frac{\boldsymbol{x}^\top \boldsymbol{C} \boldsymbol{x}}{2\left(\sum_{i=1}^d \boldsymbol{x}_i\right)^2} + \lambda \left(\min\left\{\frac{\sum_{i=1}^d \boldsymbol{m}_i \boldsymbol{x}_i}{\sum_{i=1}^d \boldsymbol{x}_i} - r, 0\right\}\right)^2 \quad \text{s.t.} \quad \|\boldsymbol{x}\|_0 \leq k$$

We use three datasets: port3, port4 and port5 from the OR-library [3], of respective dimensions $d = 89; 98; 225$. We keep $r$ and $\lambda$ the same for the 4 algorithms: $r = 0.1$, $\lambda = 10$ (for port3 and port4); and $r = 1e - 3$, $\lambda = 1e - 3$ for port5. For SZOHT, we set $k = 10$, $s_2 = 10$, $q = 10$, and $(\mu, \eta) = (0.015, 0.015)$ for port4, and $(\mu, \eta) = (0.1, 1)$ for port5 ($\mu$ and $\eta$ are both obtained by grid search over the interval $[10^{-3}, 10^3]$). For all other algorithms, we got the optimal hyper-parameters through grid search. We present our results in Figure 3.

**Few pixels adversarial attacks** We consider the problem of adversarial attacks with a sparse constraint. Our goal is to minimize $\min_\delta f(\boldsymbol{x}+\boldsymbol{\delta})$ such that $\|\boldsymbol{\delta}\|_0 \leq k$, where $f$ is the Carlini-Wagner cost function [9], that is computed from the outputs of a pre-trained model on the corresponding dataset. We consider three different datasets for the attacks: MNIST, CIFAR, and Imagenet, of dimension respectively $d = 784; 3072; 268203$. All algorithms are initialized with $\boldsymbol{\delta} = \mathbf{0}$. We set the hyperparameters of SZOHT as follows: MNIST: $k = 20$, $s_2 = 100$, $q = 100$, $\mu = 0.3$, $\eta = 1$; CIFAR: $k = 60$, $s_2 = 100$, $q = 1000$, $\mu = 1e-3$, $\eta = 0.01$; ImageNet: $k = 100000$, $s_2 = 1000$, $q = 100$, $\mu = 0.01$, $\eta = 0.015$. We present our results in Figure 4. All experiments are conducted in the workstation with four NVIDIA RTX A6000 GPUs, and take about one day to run.

## 5.4 Results and Discussion

We can observe from Figures 3 and 4 that the performance of SZOHT is comparable or better than the other algorithms. This can be explained by the fact that SZOHT has a linear convergence, but the query complexity of ZSCG and RSPGF is in $\mathcal{O}(1/T)$. We can also notice that RSPGF is faster than ZSCG, which is natural since proximal algorithms are faster than Frank-Wolfe algorithms (indeed, in case of possible strong-convexity, vanilla Frank-Wolfe algorithms maintain a $\mathcal{O}(1/T)$ rate [13], when proximal algorithms get a linear rate [4, Theorem 10.29]). Finally, it appears that the convergence of ZORO is sometimes slower, particularly at the early stage of training, which may come from the fact that ZORO assumes sparse gradients, which is not necessarily verified in real-world use cases like the ones we consider; in those cases where the gradient is not sparse, it is possible that the sparse gradient reconstruction step of ZORO does not work well. This motivates even further the need to consider algorithms able to work without those assumptions, such as SZOHT.

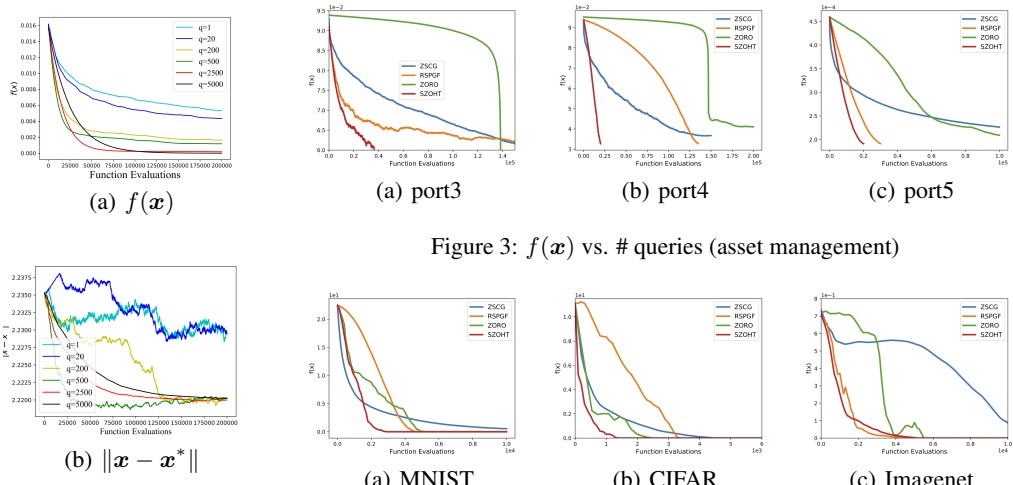

(a) $f(\boldsymbol{x})$

(a) port3     (b) port4     (c) port5

Figure 3: $f(\boldsymbol{x})$ vs. # queries (asset management)

(b) $\|\boldsymbol{x} - \boldsymbol{x}^*\|$

(a) MNIST     (b) CIFAR     (c) Imagenet

Figure 2: Sensitivity analysis

Figure 4: $f(\boldsymbol{x})$ vs. # queries (adversarial attack)

## 6 Conclusion

In this paper, we proposed a new algorithm, SZOHT, for sparse zeroth-order optimization. We gave its convergence analysis and showed that it is dimension independent in the smooth case, and weak dimension-dependent in the RSS case. We further verified experimentally the efficiency of SZOHT in several settings. Moreover, throughout the paper, we showed how the condition number of $f$ as well as the gradient error have an important impact on the convergence of SZOHT. As such, it would be interesting to study whether we can improve the query complexity by regularizing $f$, by using an adaptive learning rate or acceleration methods, or by using recent variance reduction techniques. Finally, it would also be interesting to extend this work to a broader family of sparse structures, such as low-rank approximations or graph sparsity. We leave this for future work.

## Acknowledgments and Disclosure of Funding

Xiao-Tong Yuan is funded in part by the National Key Research and Development Program of China under Grant No. 2018AAA0100400, and in part by the Natural Science Foundation of China (NSFC) under Grant No.U21B2049, No.61876090 and No.61936005.

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
