# OpenReview forum: "Zeroth-Order Hard-Thresholding: Gradient Error vs. Expansivity"
_NeurIPS.cc/2022/Conference — NeurIPS 2022 Accept_

### Official Review · Reviewer_skyg · 2022-07-07

**Rating:** 7
**Confidence:** 3
**Soundness:** 3 good
**Presentation:** 3 good
**Contribution:** 3 good

**Summary:**

This paper considers the $\ell_0$ constrained stochastic optimization problem with zeroth-order oracles only. They proposed a stochastic zeroth-order hard-thresholding algorithm, making use of a novel random support sampling. They provide the convergence guarantees of SZOHT under standard assumptions, and the query complexity of SZOHT has weak/no dependence under different settings.

**Questions:**

See "Strengths and Weaknesses"

**Ethics Review Area:**

["I don’t know"]

**Limitations:**

See "Strengths and Weaknesses"

**Strengths And Weaknesses:**

The paper proposed a new algorithm SZOHT and show that the oracle complexity is dimension independent under some condition without additional gradient sparsity assumption. The ideas and the proposed algorithm are easy to follow. The paper is well organized. The experiments show improvements over previous work. In general, the paper is good.

I have a few comments/questions:

Major Comments:
1. $k$ and $k^*$: I have two questions about $k^*$ and $k$. Based on my understanding, $k^*$ is the actual target sparsity, while $k$ is just a parameter in the algorithm, and each iterate of algorithm maintains $k$-sparsity instead of $k^*$-sparsity. That means each iterate of the algorithm does NOT maintain primal feasibility, is that correct? If that is the case, is there any polishing steps you can do to refine the solution so that it has exact $k^*$ nonzeros? The second question about this is in the algorithm, it says $k=\mathcal{O}(\kappa^4k^*)$, and also in Remark 4, the authors mention that $q$ should be taken so that $k\leq d$. But even with $q=\infty$, $\epsilon_F=2$, there are still chances that $k=\mathcal{O}(\kappa^4k^*)$ will be always greater than $d$, which means the algorithm has limitation on actual target sparsity $k^*$?
2. $\mu$: In the convergence analysis, there is only a linear term in $\mu$ while neither other constants or nor other terms depend on $\mu$. This is a bit counter-intuitive, because that means $\mu$ can be taken arbitrarily small to reduce the error term? Is there any theory to provide guidelines on how to choose $\mu$? What did you choose for $\mu$ in the numerical experiments?
3. Condition number $\kappa$ and $s_2$: in theory, the convergence guarantee and oracle complexity bound depend heavily upon the condition number $\kappa$, which is unknown or hard to compute based on its definition. The convergence guarantee may not hold if the learning rate $\eta$ and/or $q$ are chosen arbitrarily. In practice, one may need to tune heavily on these hyperparams, which the authors did for their experiments. I am wondering if there is any way you can estimate these parameters or any way to avoid large grid of hyperparam tuning? Also there any theory to provide guidelines on how to choose $s_2$? Because $s_2$ might also affect the condition number?

Minor Comments
1. In line 75/94, $q$ is not introduced yet, so it is better to give some descriptions on $q$.
2. In Proposition 1 Theorem 1, Corollary 1 and Corollary 2, the constants are hard to locate. Probably it’s better to have equation number and refer to these equations later on to remind people what these constants are in the theory.

---

> ### Author Response · Authors · 2022-08-02
> **Rebuttal for reviewer 3**
>
> Thank you for your detailed review and positive feedback.
>
> **Q1:**
>
> Yes, the iterates are only $k$ sparse, with $k\geq k^\*$, which is common in $\ell_0$ optimization (see also *About $k$ and $k^\*$* in the general rebuttal), though it might be possible to exactly recover the support $x^\* = \arg\min_{x \text{s.t.} \|\|x\|\|_0 \leq k^\*} f(x)$, under some additional conditions on $f$, as described for instance by [10], in the first-order setting: in that case, once the optimal $k^\*$-sparse support is identified, one would optimize over that support, maintaining primal feasibility (but we leave that for future work).
>
> The question about polishing steps is interesting: one could maybe use the following heuristic: after the HT algorithm has returned $x^{(T)}$, one would then use say a LASSO algorithm-type (like RSPGF [11]), *but restricting the optimization to the support of* $x^{(T)}$, (and with a grid-search to obtain an exactly $k^\*$-sparse output). Since the query complexity of RSPGF is linear in the dimension of the problem (see Table 1), that final step would just have query complexity $\mathcal{O}(k)$ instead of $\mathcal{O}(d)$. But we leave a more thorough answer to that question for future work.
>
> Regarding the "limitation on actual target sparsity $k^\*$": indeed, even if say $q=\infty$, if the problem is very ill-conditioned (that is, $\kappa \gg 1$), it can be impossible to find $k \leq d$ AND $k^\* \geq 1$ such that we have $k=\mathcal{O}(\kappa^4 k^\*)$. However, we typically consider here the very high dimensional setting (which is the main setting for sparsity constrained algorithms), in which $d \gg k$, so such problems are less likely to happen (also, this is limitation is common in $\ell\_0$ optimization).
>
> **Q2:**
>
> That is indeed an important point: yes, in theory, $\mu$ should be taken as small as possible. However, in practice, $\mu$ should be tuned, as it must be large enough to avoid any numerical precision errors. Certain theoretical recommendations for choosing $\mu$ can be found in the literature (for instance $\mu = \mathcal{O}(1/\sqrt{d T})$, or other variants from Table 1 in [8]). However, these values are typically chosen so that the convergence rate has no system error, and we believe those values given in [8]  may sometimes be misleading, since for high $d$ or $T$, $\mu=\mathcal{O}(1/\sqrt{d T})$ may be too small to use in practice due to numerical errors.
>
> In our case, since we already have a system error term with $\sigma$, it would not be possible to find a value for $\mu$ to obtain no system error overall. This is why we chose that form of result, which we find might be a bit clearer, i.e. a convergence rate with two system errors, one in terms of $\sigma$, and one in terms of $\mu$. (This form of result that keeps the $\mu$ free as a system error also appears in other papers, for instance in the seminal work of [9] (e.g. in Theorem 8)). For the experiments, we chose $\mu$ through grid search. Specifically: for the sensitivity analysis, we chose $\mu=1e-4$; for the adversarial attacks we chose $\mu=0.3$ (MNIST), $\mu=1e-3$ (CIFAR), and $\mu=0.01$ (Imagenet); for the asset management, we chose $\mu=0.015$ (port4) and $\mu=0.1$ (port5). We now have included those specific values in the new revision.
>
> **Q3:**
>
> Yes, that is a very important point. To avoid tuning the hyperparameters, we believe it could indeed be possible to use methods like backtracking line-search, which seeks a local estimate of the smoothness $L$ to choose the learning rate. This high dependence on $\kappa$ is also why we believe techniques that reduce the dependence on $\kappa$ would make our SZOHT technique even better, such as acceleration methods, as described in the Conclusion. However, we leave this for future work.
>
> Regarding the choice of $s_2$, that is indeed a very interesting question, since in theory, a given function $f$ that is $L\_{s\_2}, s\_2$-RSS is also $L\_{s'\_2}, s'\_2$-RSS (with $s'\_2 \leq s\_2$ and $L\_{s'\_2} \leq L\_{s\_2}$) (simply because the upper bound in the definition of $L\_{s\_2}, s\_2$-RSS also applies to a smaller sets of $x$, possibly with a smaller constant). So one might be tempted to decrease $s\_2$ to get a better condition number $\kappa$. But decreasing $s\_2$ increases the gradient estimation error, as per Proposition 1. So it is unclear in general how changing $s_2$ will impact the convergence overall. Generally speaking, though, we can say that if the $s\_2$-restricted smoothness constant $L\_{s\_2}$ does not depend a lot on $s\_2$, then it is better to increase $s\_2$ (to decrease the error in the gradient estimate). But if that constant decreases a lot when $s\_2$ decreases, then it can be worthy to decrease $s\_2$.
>
> **Minor Comments:**
>
> Thank you for your suggestions, we have updated the manuscript accordingly (see section **Modifications** from the general rebuttal).

---

> > ### Comment · Reviewer_skyg · 2022-08-08
> > **Thank you for the response**
> >
> > Dear authors,
> >
> > Thank you for the detailed response for my clarification questions. The response makes sense to me. I have increased the score to 7.
> >
> > Best, Reviewer

---

> > > ### Author Response · Authors · 2022-08-09
> > > **Thanks to reviewer 3**
> > >
> > > Thank you very much for your feedback and improved score.

---

### Official Review · Reviewer_bBM4 · 2022-07-10

**Rating:** 8
**Confidence:** 5
**Soundness:** 3 good
**Presentation:** 3 good
**Contribution:** 4 excellent

**Summary:**

This paper describes a zeroth-order version of the hard-thresholding algorithm. Interestingly, the convergence rate (up to system errors) is dimension-independent in the case of smooth functions and full support sampling for the random directions. The paper also provides convergence results in the more general case of restricted smoothness and restricted support for the sampling of random directions. It also discusses an interesting trade-off that arises between the error of the zeroth-order estimate, and the expansivity of the hard-thresholding operator, that results into a minimal value for the number of random directions q. Finally, the authors demonstrate the utility of their method with encouraging  experimental results on synthetic datasets such as a quadratic function, and real-life datasets such as asset management and black-box adversarial attacks.

**Questions:**

I believe one aspect that may deserve further discussion is about the system error in Theorem 1, that depends on the norm of the gradient at optimum ||nabla f(x*)||. If this term is large, the upper bound for the convergence rate might become large too. However, this term seems to appear also in other works on hard-thresholding like [2], so I guess it is not really an issue of the work itself, more of an issue intrinsic to the hard-thresholding method and l0 constraint. But I think it might be interesting to elaborate on how large this term can get in practice, and also in which settings it can be small (for instance, it may be interesting to add references on compressed sensing, in which the original parameter vector is assumed sparse, or to related assumptions like assumption (2) in [3], in which it is stated that "the existence of very sparse minimizers is a known property in over-parametrized DNNs").

**Limitations:**

Since this algorithm describes a general method for optimization, I don't believe there is any direct negative societal impact.

**Strengths And Weaknesses:**

## Originality:
The main difference compared to previous work is in tackling both zeroth-order and l0 optimization, which has not been done before (except by the work of [1], but they make stronger assumptions (sparsity of gradients)). The proof for the dimension independence/insensitiveness of zeroth-order hard-thresholding, a consequence of the bounds on the error of the ZO gradient, is novel, and uses interesting results about the surface integration on a sphere, as well as involved combinatorics computations to generalize them to the random support technique. I think it is an interesting contribution to the litterature on ZO optimization. The discussion about the minimal value for q is also very interesting and novel.

## Quality:
The authors provide detailed self-contained proofs for the main claims in the appendix. They also provide some code to run SZOHT on the toy experiments, as well as an asset management problem and an adversarial attack problem.

## Clarity:
The main results and theorems are discussed in details. Figure 1 convey very well the main message regarding the conflict between expansivity and gradient error.
I believe there are a few typos/missing notations, that could easily be addressed:
- In table 1, note 4 does not appear
- In table 1, I think FO was not defined (though I guess it means "First-order")
- In section 5.1 sensitivity analysis, I think it should be an elementwise product a  * (x - b), not a dot product (otherwise the squared norm would be just a squared number)

## Significance :
As zeroth-order optimization is known to suffer from the curse of dimensionality, as described in the introduction of the paper, dimension-insensitive results like the one from this paper are very interesting. In addition, that paper manages to achieve such results with only mild assumptions (namely, RSC and RSS, and without any gradient sparsity assumption). The experiments are extensive, and ran on real-life datasets like adversarial attacks which shows the method also works well in large scale practical machine learning problems.


# References
[1]   "Zeroth-order (non)-convex stochastic optimization via conditional gradient and gradient updates"
  Balasubramanian, Krishnakumar and Ghadimi, Saeed
  Proceedings of the 32nd International Conference on Neural Information Processing Systems 2018


[2] "Linear convergence of stochastic iterative greedy algorithms with sparse constraints",
  Nguyen, Nam and Needell, Deanna and Woolf, Tina
  IEEE Transactions on Information Theory, 2017,

[3] Ac/dc: Alternating compressed/decompressed training of deep neural networks},
  Peste, Alexandra and Iofinova, Eugenia and Vladu, Adrian and Alistarh, Dan},
  Advances in Neural Information Processing Systems
  2021

---

> ### Author Response · Authors · 2022-08-02
> **Rebuttal for reviewer 2**
>
>
> Thank you for your insightful review and appreciation of our work.
>
> > I believe there are a few typos/missing notations, that could easily be addressed:
> > In table 1, note 4 does not appear
>
> Thank you for pointing that out, we have deleted the footnote in the new revision.
>
> > In table 1, I think FO was not defined (though I guess it means "First-order")
>
> Thank you for pointing that out, we have updated the manuscript to define FO in the legend of the Table.
>
> > In section 5.1 sensitivity analysis, I think it should be an elementwise product a * (x - b), not a dot product (otherwise the squared norm would be just a squared number)
>
> Thank you for pointing that out, it should indeed have been an elementwise product, we have updated the manuscript.
>
>
> > I believe one aspect that may deserve further discussion is about the system error in Theorem 1, that depends on the norm of the gradient at optimum ||nabla f(x*)||. If this term is large, the upper bound for the convergence rate might become large too. However, this term seems to appear also in other works on hard-thresholding like [2], so I guess it is not really an issue of the work itself, more of an issue intrinsic to the hard-thresholding method and l0 constraint. But I think it might be interesting to elaborate on how large this term can get in practice, and also in which settings it can be small (for instance, it may be interesting to add references on compressed sensing, in which the original parameter vector is assumed sparse, or to related assumptions like assumption (2) in [3], in which it is stated that "the existence of very sparse minimizers is a known property in over-parametrized DNNs").
>
> This is indeed an interesting point:  the term in $\sigma$ from Theorem 1 indeed depends on $\|\|\nabla f(x^\*)\|\|$, which could indeed be large depending on the problem. However, as you mentioned, this is a classical issue in $\ell_0$ optimization and hard-thresholding-type algorithms, and there exist many problems for which that term can be considered small enough, as the ones you mentioned. In the new revision, we have included in Remark 2 a discussion including the references you mentioned, about settings in which the term $\|\|\nabla f(x^\*)\|\|$ is considered small.

---

> ### Comment · Reviewer_bBM4 · 2022-08-08
> **Reply**
>
> Hi,
> Your rebuttal has addressed all of my main concerns!

---

> > ### Author Response · Authors · 2022-08-09
> > **Thanks to reviewer 2**
> >
> > Thank you very much for your feedback and improved score.

---

### Official Review · Reviewer_baZg · 2022-07-14

**Rating:** 6
**Confidence:** 3
**Soundness:** 3 good
**Presentation:** 3 good
**Contribution:** 3 good

**Summary:**

This work considers the problem of stochastic optimization in the presence of an L0 ball constraint (i.e. at most k components of the minimizer can be nonzero). This is essentially stochastic optimization in a setting where a sparse solution is sought. In particular, the authors consider this problem in a zeroth-order (ZO) optimization situation where (stochastic) gradients are not available, useful for black-box adversarial attacks and reinforcement learning (for example).

Specifically, the paper proposes to use iterative hard thresholding, a standard method for L0-constrained problems, but with stochastic gradients estimated via gradient smoothing (a form of randomized finite differencing). The gradient smoothing is specifically designed to apply sparse perturbations to the current iterate, unlike standard smoothing.

They prove a linear convergence rate for their algorithm to a neighborhood of a minimizer under some restricted versions of strong convexity and L-smoothness. The neighborhood of convergence depends on both the variance of the stochastic gradients, and the smoothing parameter used in the gradient estimator. Under full L-smoothness of the objective, they prove a fully dimension-independent convergence rate, which improves on existing sparse ZO optimization based on L1 regularization. A key restriction of this technique is that the convergence theory requires the number of gradient smoothing estimates to be sufficiently large (to ensure the estimated stochastic gradient is sufficiently close to the true stochastic gradient).

The authors provide numerical experiments for sparse portfolio optimization and adversarial example generation, showing that their proposed method outperforms existing sparse ZO optimization methods.

**Questions:**

1. In the gradient estimator eq (2), is it correct that each u_i has a different support set? In that case, the gradient estimator could have up to q*s_2 nonzero entries. Is this a concern from a computational effort point of view?

2. Throughout Section 4, what is k*? Is this just an upper bound on ||x_0||_0 (in which case if x_0=0 then any k* is acceptable), or something else?

3. The conditions of Theorem 1, in particular $s \leq d$ implies that we must have $k\leq d/2$. Does this cause any conflict with the other assumption $k \geq k^* \rho^2 / (1-\rho^2)^2$? Any restrictions on the problem constant $k$ should be explicitly given (even though $k\leq d/2$ is not a problematic requirement).

4. It is mentioned in line 219 that Lemma 1 gives conditions for which $\rho \gamma < 1$. However, the lemma's statement does not appear to include this. What specifically does $q \geq q_{\min}$ imply? In particular, does it imply either/both of $k \geq k^* \rho^2 / (1-\rho^2)^2$ or $\rho\gamma<1$?

5. Figure 2b seems to show that there is essentially no reduction of $\|x-x^*\|$ for any value of $q$. Does this mean your choice of problem makes the non-decaying ($\sigma$ and $\mu$) terms in Theorem 1 too large to show useful results in this regard?

**Limitations:**

The main limitations are related to problem assumptions and requirements on $q$ and $k$. These are reasonably well described, noting my feedback above. There is no concern about negative societal impact from this work.

**Strengths And Weaknesses:**

To my knowledge, this is the first attempt to combine hard thresholding with ZO gradient estimators (although it has already been applied to stochastic gradients). The resulting algorithm is simple and appears to perform well in practice. I note that in practice both the learning rate and the smoothing parameter have to be tuned, which is expensive, but a reasonable cost in this setting. The convergence theory is interesting and the implications for the gradient smoothing accuracy will likely affect any future L0-constrained ZO algorithm development.

Overall I find this an interesting work. The quality of prose is sufficiently high in most places.

My main concern is the discussion around the choice of $q$ (particularly Lemma 1) in Section 4. My understanding of Lemma 1 is that first a suitable $q \geq q_{\min}$ is chosen, and that then defines a valid range of $k$. This seems to be the wrong way around - usually we would have a $k$ specified by the problem, and then choose how to construct the ZO gradient estimators with a corresponding restriction on $q$ (depending on $k$). It also appears that the authors require, if the chosen $k$ is too small (although exactly what this means is unclear as per my concern around Lemma 1) that a larger $k$ is chosen. That is, the actual L0 constraint must be changed. This is quite unusual, in that an optimization algorithm requires the solution of a *different* problem than originally specified.

Minor comments:
- What is footnote 4 in Table 1 (for SZOHT in the second-last row)?
- The variable $q$ is mentioned several times in the introduction, but not defined until later.
- I find Figure 1 extremely unclear and don't think it contributes to the text (lines 174-178). I recommend it be removed.
- Line 298, "proximal algorithms are faster than Frank-Wolfe algorithms". I have not seen this conclusion before (as a non-expert in these methods) - do you have a citation for this?

---

> ### Author Response · Authors · 2022-08-02
> **Rebuttal for reviewer 1**
>
> Thank you for your detailed review and positive feedback.
>
> >  usually we would [...] choose [...] $q$ (depending on $k$).
>
> Yes, we should choose $q$ in terms of $k$: a possible way to do so appears in Cor. 1 (and 2). The goal of Lem. 1 is rather just to warn the reader, that there is a necessary minimal value of $q$ for Theorem 1, which is unusual in ZO optim. We elaborate on this in the new revision, and since Lem. 1 is secondary (compared to Cor. 1 and Cor. 2), we rewrote it as a Remark.
>
> > It also appears that [...] if the chosen $k$ is too small [...] a larger $k$ is chosen. That is, the actual L0 constraint must be changed.
>
> Yes, if $k$ is too small (i.e. $< \rho^2 k^\*/ (1 - \rho^2)^2$), then $\rho \gamma > 1$, so convergence will not be guaranteed. We agree this could be seen as "changing" the problem, but we would rather say "we solve a (fixed) $k^\*$ sparse problem, but with a more relaxed sparsity $k$":  this is common in the $\ell\_0$ optimization literature (see "About $k$ and $k^\*$" from the global rebuttal).
>
>
> > Minor comments:
>
> Thanks for your suggestions, we have fixed the minor points in the new revision.
>
> Regarding Fig. 1, we realized that l. 213-217 may correspond more to it, so we added the ref. to Fig. 1 there instead, but if the figure is still unclear we can remove it.
>
> Regarding Frank-Wolfe vs. proximal GD, the reason is that vanilla Frank-Wolfe cannot exploit strong-convexity [6, Sec. 1.1, 2nd §], contrary to proximal GD (which has a linear rate under strong-convexity [7, Theorem 10.29]) (even if the $f$ is not globally strongly convex, proximal GD could still exploit strong-convexity if present locally). We have included a short sentence about that in the new revision, thanks for pointing it out.
>
>
> > Questions:
> > [...] is it correct that each $u_i$ has a different support set? [...] Is this a concern from a computational effort point of view?
>
> It is correct: we could have considered instead a fixed support for all $u_i$, but this would only estimate the gradient along that fixed block, and our convergence analysis would need modifications (it would resemble the analysis of random block-coordinate descent [12], which is not the scope of our paper). Rather, we want our estimator to tend to $\nabla f_{\mu}$ when $q$ increases (which is not the case in [12]), to easily bound $\mathbb{E} \|\|\hat{\nabla} f(x) - \nabla f(x)\|\|$, which we need in our convergence analysis.
> This could indeed require a $q\* s_2$ size vector, but it could still be efficient say in the distributed setting, where each of $q$ edge devices, memory limited, would just need to form an $s_2$-size random vector, and only the centralized server would form the $q\* s_2$-sized vector. Thanks for that remark, we included part of this answer in the new revision l. 132-134.
>
>
> > What is $k^\*$? [...] an upper bound on ||x\_0||\_0 [...] ?
>
> Actually, $k^\*$ is not an upper bound on $\|\|x\_0\|\|\_0$, but is the sparsity of the minimizer $x^\* = \arg\min\_{x \text{s.t.} \|\|x\|\|\_0 \leq k^\*}$ which we can guarantee convergence to (up to system error): we do not guarantee convergence to a $k$ sparse minimizer of $f$, but to a sparser ( $k^\*$ -sparse) minimizer instead (see "About $k$ and $k^\*$" in the global rebuttal above).
>
> > [...] we must have $k\leq d/2$. Does this cause any conflict with the other assumption $k\geq k^\* \rho^2/(1 - \rho^2)^2$?
>
> Yes, if the problem is ill-conditioned ( $\kappa >> 1$ ), and $d$ is not very large, those two assumptions can be in conflict. But typically, our work targets the high-dimensional setting (as most sparsity-based methods), so this should ensure $k\leq d/2$. (Such kinds of conditions are also present in references from section "About $k$ and $k^\*$" of the general rebuttal above).
> Writing explicitly both bounds would indeed be better: we added that to the new revision.
>
> > [...] Lemma 1 gives conditions for which $\rho\gamma<1$. However, the lemma's statement does not appear to include this. What specifically does $q\geq q_{\text{min}}$ imply? [...] ?
>
> Thanks, $\rho \gamma < 1$ is indeed a consequence of $k >  k^\* \frac{\rho^2}{(1- \rho^2)^2}$. We have added that in the revision of Lemma 1.
>
> > Figure 2b [...] there is [...] no reduction of $|x - x^\*|$  for any value of $q$. Does [...] your choice of problem makes the non-decaying ($\sigma$ and $\mu$) terms in Theorem 1 too large to show useful results [...]?
>
> Indeed, for any $k^\* \leq k$, we had  $\arg \min\_{x \~ \text{s.t.} \|\|x\|\|\_0 \leq k^\* } f(x)  \ni x^\* := \Phi_{k^\*} (b)$ , (with $\Phi\_{k^\*}$ the hard-thresholding operator keeping the $k^\*$ largest components (in magnitude)), for which we had $\|\|\nabla f(x^\*)\|\| = a * \|\|b - \Phi_{k^\*}(b )\|\| \neq 0$. We updated $f$ so that $\sigma$ is smaller, and we can now clearly see that $\|\|x - x^\*\|\|$ decreases for large $q$ , and increases for small $q$.

---

### Author Response · Authors · 2022-08-02
**Global rebuttal**

We sincerely thank all the reviewers for their insightful review and appreciation of our work. We highlight below the main remarks, and the modifications we made to our manuscript according to the reviews.

Overall, the reviewers agreed that this work is novel, in combining both zeroth-order and hard-thresholding under mild assumptions, that the dimension (weak)/(in)dependence, without any sparsity assumption on the gradient, improves upon previous work, theoretically and experimentally, and that this "will likely affect any future L0 constrained ZO algorithm development".


- __About $k$ and $k^\*$__ Many concerns of the reviewers were regarding the conditions for the main results to apply, and we give here some general clarifications regarding this question. Since in general, $\ell\_0$-constrained optimization is NP-hard to solve (even approximately)[1], many results in the hard-thresholding literature give a convergence rate (up to system error) for $k$-sparse iterates $x^{(t)}$, in terms of $\|\|x^{(t)} - x^\*|\|$, where $x^\*=\arg\min\_{x \~ \text{s.t.} \~ \|\|x\|\|\_0 \leq k^\*}$ with $k^\* \ll k$. That is, they provide convergence rates for a *relaxed sparsity* $k$ of the iterates ([2, 3, 4]). This explains why we also have $k$ and $k^\*$ appearing in our results: this is actually intrinsic to $\ell\_0$ optimization. What we believe is new in our work, however, is an analysis of the interaction between the number of random directions $q$ in ZO optimization, and the variables specific to $\ell\_0$ optimization ($k$, $k^\*$, and $\kappa$). Specifically, we showed how to choose values of $q$ (and other parameters such as the learning rate $\eta$) such that convergence is guaranteed (up to system error).


- **Modifications:**  Following the reviewers' suggestions, we have included the following modifications in the new revision: We:
  - Improved presentation and typos (including: removed footnote 4, defined $q$ in Introduction, referenced the constants in our results)
  - Clarified Theorem 1 by making the upper bound on $k$ explicit, in addition to the lower bound
  - Highlighted the advantage of our random support sampling in the distributed setting, (lines 132-134)
  - Clarified Lemma 1, rewrote it as a Remark, and gave a more informative condition in the case $s_2=1$
  - Modified our sensitivity analysis to showcase iterates $x^{(t)}$ getting closer to $x^\*$ for large $q$, and further away for small $q$.
  - Included an additional sentence about settings where $\|\|\nabla f(x^\*)\|\|=0$, in Remark 2
  - Added a sentence to elaborate on why proximal algorithms are generally faster than Frank-Wolfe, in paragraph 5.4.


**References for the individual rebuttals:**

[1] Y. Chen and M. Wang. Hardness of approximation for sparse optimization with L0 norm.
Technical Report, 2016.


[2] On Iterative Hard Thresholding Methods for High-dimensional M-Estimation
 Jain, Prateek and Tewari, Ambuj and Kar, Purushottam,
 Advances in Neural Information Processing Systems, 2014


[3] Efficient Stochastic Gradient Hard Thresholding
    Zhou, Pan and Yuan, Xiaotong and Feng, Jiashi,
    Advances in Neural Information Processing Systems, 2018


[4] Iterative Hard Thresholding with Adaptive Regularization: Sparser Solutions Without Sacrificing Runtime,
    Axiotis, Kyriakos and Sviridenko, Maxim,
    Proceedings of the 39th International Conference on Machine Learning, 2022

[5] Efficient Stochastic Gradient Hard Thresholding
    Zhou, Pan and Yuan, Xiaotong and Feng, Jiashi,
    Advances in Neural Information Processing Systems, 2018

[6] Faster rates for the frank-wolfe method over strongly-convex sets,
  Garber, Dan and Hazan, Elad,
  International Conference on Machine Learning, 541--549, 2015

[7] First-order methods in optimization,
	Beck, Amir,
    2017, SIAM

[8]  A primer on zeroth-order optimization in signal processing and machine learning: Principals, recent advances, and applications
  Liu, Sijia and Chen, Pin-Yu and Kailkhura, Bhavya and Zhang, Gaoyuan and Hero III, Alfred O and Varshney, Pramod K,
  IEEE Signal Processing Magazine, volume 37, pages 43--54, 2020


[9] Random gradient-free minimization of convex functions
   Nesterov, Yurii and Spokoiny, Vladimir,
  Foundations of Computational Mathematics,
  17, 527--566, 2017,

[10] Exact recovery of hard thresholding pursuit,
    Yuan, Xiaotong and Li, Ping and Zhang, Tong,
    Advances in Neural Information Processing Systems, 2016

[11] Mini-batch stochastic approximation methods for nonconvex stochastic composite optimization.
    Saeed Ghadimi, Guanghui Lan, and Hongchao Zhang. Mathematical Programming, 2016.

[12] A comprehensive linear speedup analysis for asynchronous stochastic parallel optimization from zeroth-order to first-order.
     Xiangru Lian, Huan Zhang, Cho-Jui Hsieh, Yijun Huang, and Ji Liu.,
     Advances in Neural Information Processing Systems, 2016.

---

### Meta-Review · Area_Chair_FDqv · 2022-08-21

**Recommendation:** Accept
**Confidence:** Certain

**Metareview:**

The paper considers stochastic optimization in the presence of an L0 ball constraint. All reviewers agree that the theoretical derivations are solid and numerical experiments show promising performance. Although the algorithm might have to be finely tuned to perform well, it contains many novel and interesting elements that warrants its publication.

**Award:**

No

---

### Decision · Program_Chairs · 2022-09-14

Accept